# Storage and disposal practices of unused and expired medications among adult women attending Banadir Hospital, Mogadishu, Somalia: A cross-sectional study

Abdiweli Mohamed Abdi[1]*, Osman Mohamed Mohamud[2], Mohamed Ismail Iman[3], Nor Haji Osman[4], Ahmed Abdulkadir Kalif[5], Mahad Ali Mohamud[6], Abdinor Hussein Ahmed[7], Mohamed Yusuf Isak[8]

1 Department of Public Health, Faculty of Health Sciences, Al Hayat Medical University, Mogadishu, Somalia, 2 Department of Medical Laboratory, Faculty of Health Sciences, Al Hayat Medical University, Mogadishu, Somalia, 3 Department of Medicine, Faculty of Medicine and Surgery, Al Hayat Medical University, Mogadishu, Somalia, 4 Directorate of Health and Human Services, Banadir Regional Administration, Mogadishu, Somalia, 5 Department of Public Health, Federal Ministry of Health and Human Services, Mogadishu, Somalia, 6 Department of Public Health, Somali Regional Health Bureau, Jigjiga, Ethiopia, 7 Department of Pharmacovigilance, Federal Ministry of Health and Human Services, Mogadishu, Somalia, 8 Department of Public Health, Faculty of Health Sciences, Mogadishu University, Mogadishu, Somalia

* cabdiwalishiikh@gmail.com

## Abstract

Improper storage and disposal of unused and expired medications in households pose significant environmental and public health risks. A cross-sectional study was conducted in November 2025 among 427 consecutively selected women visiting the outpatient pharmacy of Banadir Hospital to assess storage and disposal practices of unused and expired medications. Data were analyzed using descriptive statistics, while associations between socio-demographic characteristics and medication storage and disposal practices were assessed using the chi-square test and interpreted with Cramér's V. Most respondents reported storing unused medications at home (97%), mainly for emergency situations, while more than one-third also kept expired medications at home (34%). Nearly all participants did not read storage instructions on medication labels (98%), and most did not check expiration dates before purchasing (70%) or using medications (74%). Disposal practices were inappropriate in most cases, with over 90% reporting disposal in household trash. These findings indicate that improper medication storage and disposal practices are highly prevalent in this setting. Public awareness programs and the establishment of a national drug take-back system are needed to reduce environmental contamination, prevent medication misuse, and improve public health safety.

**Data availability statement:** All relevant data are within the paper and its Supporting Information files.

**Funding:** The author(s) received no specific funding for this work.

**Competing interests:** The authors have declared that no competing interests exist.

## Introduction

Medications play a crucial role in addressing human health problems; however, proper storage and disposal of unused and expired medications are equally important [1].

Proper medication storage practice is defined as storing medications in accordance with the manufacturer's labeled storage conditions and keeping them out of the sight and reach of children, whereas proper medication disposal practice is defined as the use of drug take-back or return options [1–3].

The combined annual amount of prescription and over-the-counter (OTC) medications exceeds 1,000,000 tons [4]. However, patients might not take all of the medications that have been given to them for a variety of reasons [5]. The primary reasons include symptom relief, dosage changes, forgetfulness, intolerance of side effects, large quantities prescribed by physicians or pharmacists, and medication expiration [6]. Additionally, some patients store their medications at home for future use [7]. Improper disposal of expired and unused medications is common, with many people flushing them down the toilet or sink, throwing them into household trash, or leaving them stored in medicine cabinets [8,9]. These practices are often driven by limited awareness and the absence of local medication disposal programs [10]. Such improper disposal poses significant risks to public health, the environment, and animal life, including accidental ingestion, misuse, and contamination of water sources [11]. Adult women are an important population for assessing the storage and disposal practices of medications, as they usually store and dispose of medications used in their households and are also more likely to obtain them due to reproductive health issues [12,13]. The collapse of the Somali central government in 1991 led to the breakdown of the national health system, including the pharmaceutical sector. However, in 2014, the Somali National Medicines Policy was established to strengthen pharmaceutical governance and regulation [14]. By November 2023, the first edition of the Somali National Pharmacovigilance Guidelines was published by the Interim National Medicine Regulatory Authority [15]. Although the Federal Ministry of Health and its partners are undertaking reforms, there are currently no national guidelines for the storage and disposal of unused and expired medications, and no national drug take-back scheme is in place. To our knowledge, this is the first study conducted in Somalia on this topic and aimed to assess the storage and disposal practices of unused and expired medications among adult women attending Banadir Hospital, Mogadishu, Somalia.

## Materials and methods

### Study design, period and setting

A quantitative institution-based cross-sectional study was carried out in November 2025 at the outpatient pharmacy of Banadir Hospital. This is a maternal and child health hospital, a referral hospital for patients from across Somalia, as well as a teaching hospital. The hospital serves more than 3,000 patients per month [16].

## Study population, criteria, and sampling

Adult women were selected for this study because they often play a central role in obtaining, storing, and disposing of medications within households. In addition, women and children are among the most vulnerable groups to disease and other health problems in Somalia [17], which may increase women's involvement in acquiring medications for family use. Women who were 18 years of age and older and visited Banadir Hospital's outpatient pharmacy to obtain prescribed medications were consecutively approached. Women who were unwilling to participate, not feeling well enough to participate, or residing outside the study area were excluded from the study.

A consecutive sampling approach was used to recruit all eligible women attending the outpatient pharmacy during the study period. This method was selected because it allows the inclusion of all accessible participants who meet the eligibility criteria in a clinical setting and is commonly used in hospital-based cross-sectional studies where a complete sampling frame is not available. The outpatient pharmacy was chosen as the recruitment site because it represents the point where patients obtain prescribed medications, making it an appropriate setting to assess medication storage and disposal practices. However, this approach may have introduced selection bias, since the sample only included adult women visiting the outpatient pharmacy and was therefore not necessarily representative of adult women visiting other departments of the hospital.

Cochran's single population proportion formula was used:

$$n = Z^2 \, p \frac{(1-p)}{d^2}$$

Where:

- n = required sample size

- Z = Z-score corresponding to the 95% confidence level (1.96)

- p = estimated proportion of the population (0.5, assuming maximum variability)

- d = margin of error (0.05)

Substituting the values into the formula yielded a minimum sample size of 384 participants. To compensate for a potential 10% non-response rate, the sample size was adjusted using the non-response correction formula:

$$n = \frac{384}{1-0.10} = 427$$

Therefore, the final sample size for this study was 427 participants.

## Data collection instrument and procedure

The questionnaire was developed based on previously published studies [6,11,18–21] and consisted of three sections: socio-demographic characteristics, storage and disposal practices of unused medications, and storage and disposal practices of expired medications. Socio-demographic characteristics were treated as independent variables, while storage and disposal practices for unused and expired medications were treated as dependent variables.

To ensure content validity, the questionnaire was reviewed by two public health experts with experience in pharmaceutical practice. The instrument was pretested among 43 respondents (approximately 10% of the final sample) who were not included in the main study to assess the clarity, relevance, and comprehensibility of the questions, and minor revisions were made based on the feedback. The internal consistency of the questionnaire was evaluated using Cronbach's alpha, which yielded a reliability coefficient of 0.790, indicating acceptable reliability [22].

Data were collected through interviewer-administered face-to-face interviews conducted by trained data collectors using the structured questionnaire. Before each interview, participants were informed about the purpose of the study, and informed consent was obtained. During the interviews, data collectors explained that "unused medication" referred to any medication that had been obtained but was not fully consumed or was no longer intended for use [2], while "expired medication" referred to any medication that had passed the expiry date indicated on its packaging or label [1]. Completed questionnaires were checked daily for completeness and consistency by the research team. The complete questionnaire used for data collection is provided in S1_File.

### Bias control

Several measures were taken to minimize potential biases common to observational studies. To minimize selection bias, consecutive sampling was employed to invite all eligible women visiting the outpatient pharmacy. To avoid measurement bias and ensure language and conceptual precision, the questionnaire was translated into the local language (Somali) and then back-translated to English by two bilingual experts. To minimize interviewer bias, the data collection team received two days of comprehensive training to administer the questionnaire consistently.

### Data analysis

Data were cleaned using Microsoft Excel version 16 and then imported into SPSS version 25 for analysis. Variables were coded and assigned value labels before analysis. Descriptive statistics were used to summarize frequencies and percentages. Continuous variables such as age were summarized using mean ± standard deviation. Associations between socio-demographic variables and medication storage and disposal practices were evaluated using the chi-square test. For statistically significant chi-square tests ($p < 0.05$), the magnitude of association was assessed using Cramér's V and interpreted as weak (≤0.20), moderate (>0.20 to ≤0.60), or strong (>0.60) [23].

### Ethics approval and consent to participate

The Research Ethics Committee (REC) of Al Hayat Medical University reviewed and approved this study (AHREC/2025/077). This study was carried out in accordance with the Helsinki Declaration of 1964 and its later amendments. All respondents provided informed verbal consent and voluntarily participated in this research. Verbal consent was used instead of written consent because many participants were reluctant to sign documents, as written agreements are commonly associated with legal issues in the local context. The study objectives and procedures were explained to each participant in the Somali language before participation. After participants verbally agreed to participate, the data collectors documented the consent by marking a designated consent checkbox on the questionnaire form. The consent process was conducted in the presence of the trained interviewer who served as a witness. The Research Ethics Committee of Al Hayat Medical University reviewed and approved the use of verbal informed consent for this study.

## Results

### Participants

A total of 481 women were approached during data collection. Of these, 54 did not meet the eligibility criteria. The remaining 427 eligible women provided verbal consent and agreed to participate in the study. The final analysis included all 427 respondents who completed the survey.

The primary reason for exclusion (n = 31) was that participants came from other regions of the country. Other reasons included unwillingness to participate (n = 11), not feeling well enough to participate (n = 8), and being under 18 years of age (n = 4).

## Socio-demographic characteristics of the participants

The mean age of the respondents was 27 ± 6 years. The majority of respondents were married (373, 87.3%). Nearly half of the participants (199, 46.6%) had no formal education. Most participants (185, 43.3%) reported a monthly income of 201–300 USD. However, 359 (84.0%) of the participants were unemployed.

## Storage of unused medications and socio-demographic characteristics

As shown in Table 1, possession of unused medications was universal among unmarried women (100%), those with lower educational attainment (100%), and participants in the lower income categories (97.6%–100%). In contrast, the lowest prevalence of unused medication possession was observed among participants in the highest income category (>400 USD) (22.2%).

## Storage of expired medications and socio-demographic characteristics

As shown in Table 1, the highest prevalence of expired medication possession was observed among participants with primary education (50.7%) and those in the lowest income category (≤100 USD) (100%). Conversely, the lowest prevalence was observed among participants in the highest income category (>400 USD) (0%) and among women with university education (4.3%).

**Table 1. Storage of unused and expired medications according to the socio-demographic characteristics of the participants (n = 427).**

| Variables | Storage of an unused medication (Yes) | Storage of an unused medication (No) | Storage of an expired medication (Yes) | Storage of an expired medication (No) | Storage of an expired medication (Don't know) |
|---|---|---|---|---|---|
| **Age group** | | | | | |
| 18–27 | 257 (96.6%) | 9 (3.4%) | 106 (39.8%) | 138 (51.9%) | 22 (8.3%) |
| 28–37 | 138 (97.9%) | 3 (2.1%) | 35 (24.8%) | 100 (70.9%) | 6 (4.3%) |
| 38–47 | 17 (94.4%) | 1 (5.6%) | 3 (16.7%) | 13 (72.2%) | 2 (11.1%) |
| >47 | 2 (100%) | 0 (0.0%) | 1 (50.0%) | 1 (50.0%) | 0 (0.0%) |
| **Marital status** | | | | | |
| Married | 360 (96.5%) | 13 (3.5%) | 127 (34.0%) | 217 (58.2%) | 29 (7.8%) |
| Unmarried | 54 (100.0%) | 0 (0.0%) | 18 (33.3%) | 35 (64.8%) | 1 (1.9%) |
| **Education** | | | | | |
| No formal education | 199 (100.0%) | 0 (0.0%) | 80 (40.2%) | 116 (58.3%) | 3 (1.5%) |
| Primary | 73 (100.0%) | 0 (0.0%) | 37 (50.7%) | 29 (39.7%) | 7 (9.6%) |
| Intermediate | 47 (100.0%) | 0 (0.0%) | 13 (27.7%) | 29 (61.7%) | 5 (10.6%) |
| Secondary | 57 (91.9%) | 5 (8.1%) | 13 (21.0%) | 38 (61.3%) | 11 (17.7%) |
| University | 38 (82.6%) | 8 (17.4%) | 2 (4.3%) | 40 (87.0%) | 4 (8.7%) |
| **Occupation** | | | | | |
| Employed | 66 (97.1%) | 2 (2.9%) | 15 (22.1%) | 47 (69.1%) | 6 (8.8%) |
| Unemployed | 348 (96.9%) | 11 (3.1%) | 130 (36.2%) | 205 (57.1%) | 24 (6.7%) |
| **Monthly income** | | | | | |
| ≤100 | 2 (100.0%) | 0 (0.0%) | 2 (100.0%) | 0 (0.0%) | 0 (0.0%) |
| 101–200 | 121 (97.6%) | 3 (2.4%) | 17 (13.7%) | 91 (73.4%) | 16 (12.9%) |
| 201–300 | 183 (98.9%) | 2 (1.1%) | 69 (37.3%) | 113 (61.1%) | 3 (1.6%) |
| 301–400 | 106 (99.1%) | 1 (0.9%) | 57 (53.3%) | 39 (36.4%) | 11 (10.3%) |
| >400 | 2 (22.2%) | 7 (77.8%) | 0 (0.0%) | 9 (100.0%) | 0 (0.0%) |

## Storage and disposal practices of unused medications

As shown in **Table 2**, the majority of participants (97%) reported storing unused medications at home, primarily for emergencies (83%). Almost all participants (98%) reported that they did not read the storage instructions provided on medication labels or leaflets. More than half of the respondents stored unused medications in bedroom cabinets (55%), and the most common disposal method for unused medications was household trash (90%).

## Storage and disposal practices of expired medications

As shown in **Table 3**, 34% of participants reported keeping expired medications at home, with the most common reason being that they forgot the medications were there (26.9%). More than two-thirds of the participants (70%) reported that they did not check the expiration date of medicines before purchasing them, and nearly three-quarters (74%) did not check the expiration date before using the medications. The majority of respondents reported being familiar with the proper methods for disposing of nearly expired (85%) and expired medications (93%). However, despite this reported awareness, the majority of participants disposed of nearly expired (90%) and expired medications (91%) in household trash.

**Table 2. Storage and disposal practices of unused medications among women attending Banadir Hospital, Mogadishu, Somalia (n = 427).**

| Items | Frequency | Percent (%) |
|---|---|---|
| **Do you currently have unused medication at home?** | | |
| Yes | 414 | 97.0 |
| No | 13 | 3.0 |
| **If yes, why do you keep unused medication? (n = 414)** | | |
| For future use | 29 | 7.0 |
| For emergency use | 344 | 83.0 |
| Share with friends or family members | 41 | 10.0 |
| **Are you familiar with proper methods for storing medications at home?** | | |
| Yes | 254 | 60.0 |
| No | 173 | 40.0 |
| **Do you read the storage directions provided on the labels or leaflets?** | | |
| Yes | 9 | 2.0 |
| No | 418 | 98.0 |
| **Where do you keep the unused medication?** | | |
| Kitchen cabinet | 24 | 6.0 |
| Bathroom cabinet | 43 | 10.0 |
| Bedroom cabinet | 238 | 55.0 |
| Medicine box | 43 | 10.0 |
| Refrigerator | 79 | 19.0 |
| **Do you dispose of unused medications?** | | |
| Yes | 265 | 62.0 |
| No | 162 | 38.0 |
| **How do you dispose of unused medications? (n = 265)** | | |
| Discard in household trash | 239 | 90.0 |
| Share with friends or family members | 15 | 6.0 |
| Return to a pharmacy | 9 | 3.0 |
| Return to a hospital or clinic | 2 | 1.0 |

**Table 3. Storage and disposal practices of expired medications among women attending Banadir Hospital, Mogadishu, Somalia (n = 427).**

| Variables | Frequency | Percent (%) |
|---|---|---|
| **Do you currently have expired medication at home?** | | |
| Yes | 145 | 34.0 |
| No | 252 | 59.0 |
| I don't know | 30 | 7.0 |
| **If yes, why do you keep expired medication? (n = 145)** | | |
| I do not have time to sort through them | 13 | 9.0 |
| I forgot they were there | 39 | 26.9 |
| I do not consider them dangerous | 21 | 14.5 |
| I was planning to dispose of them anyway | 37 | 25.5 |
| I don't know | 35 | 24.1 |
| **Do you check the expiration date before purchasing a medication?** | | |
| Yes | 128 | 30.0 |
| No | 299 | 70.0 |
| **Do you check the expiration date before using a medication?** | | |
| Yes | 111 | 26.0 |
| No | 316 | 74.0 |
| **Are you familiar with the proper method for disposing of a nearly expired medication?** | | |
| Yes | 364 | 85.0 |
| No | 63 | 15.0 |
| **What method do you consider appropriate for disposing of a nearly expired medication? (n = 364)** | | |
| Discard in household trash | 328 | 90.0 |
| Return to a hospital or clinic | 15 | 4.0 |
| Keep at home until expired | 10 | 2.8 |
| Return to a pharmacy | 7 | 2.0 |
| Flush in toilet or sink | 2 | 0.5 |
| Share with friends or family members | 2 | 0.5 |
| **Are you familiar with the proper method for disposing of an expired medication?** | | |
| Yes | 395 | 93.0 |
| No | 32 | 7.0 |
| **What is the proper method for disposing of an expired medication? (n = 395)** | | |
| Discard in household trash | 359 | 91.0 |
| No action (keep at home) | 16 | 4.0 |
| Return to a pharmacy or hospital | 12 | 3.0 |
| Flush in toilet or sink | 8 | 2.0 |

## Association between the storage and disposal practices of unused medications and the socio-demographic characteristics

As shown in Table 4, educational status was moderately associated with possession of unused medications at home ($\Phi c = 0.333$, $p < 0.001$) but weakly associated with reading the storage directions ($\Phi c = 0.165$, $p = 0.001$) and the storage location of unused medications ($\Phi c = 0.173$, $p = 0.001$). Similarly, monthly income was strongly associated with possession

**Table 4. Association between the storage and disposal practices of unused medications and the socio-demographic characteristics of the participants.**

| Storage and disposal practices of unused medications | Age group | Marital status | Education | Occupation | Monthly Income |
|---|---|---|---|---|---|
| Do you currently have unused medication at home? | Φc=0.047 p=0.622 | Φc=0.067 p=0.238 | **Φc=0.333 p<0.001*** | Φc=0.003 p=0.957 | **Φc=0.639 p<0.001*** |
| If yes, why do you keep unused medication? | Φc=0.109 p=0.225 | Φc=0.075 p=0.666 | Φc=0.154 p=0.101 | Φc=0.091 p=0.471 | Φc=0.144 p=0.301 |
| Are you familiar with proper methods for storing medications at home? | Φc=0.257 p=0.061 | Φc=0.060 p=0.818 | Φc=0.150 p=0.061 | Φc=0.075 p=0.661 | Φc=0.146 p=0.203 |
| Do you read the storage directions provided on the labels or leaflets? | Φc=0.129 p=0.123 | Φc=0.099 p=0.525 | **Φc=0.165 p = 0.001*** | Φc=0.105 p=0.451 | Φc=0.198 p=0.053 |
| Where do you keep the unused medication? | Φc=0.110 p=0.623 | Φc=0.119 p=0.413 | **Φc=0.173 p = 0.001*** | Φc=0.150 p=0.145 | Φc=0.117 p=0.495 |
| Do you dispose of unused medications? | Φc=0.098 p=0.252 | Φc=0.095 p=0.051 | **Φc=0.205 p = 0.001*** | Φc=0.050 p=0.301 | Φc=0.128 p=0.136 |
| How do you dispose of unused medications? | Φc=0.085 p=0.689 | Φc=0.127 p=0.143 | Φc=0.164 p=0.061 | Φc=0.089 p=0.496 | **Φc=0.134 p = 0.014*** |

**Note:** Chi-square test was used to assess associations. Φc represents Cramer's V effect size. ***p<0.05** indicates statistical significance.

of unused medications at home (Φc=0.639, p<0.001) but weakly associated with the disposal method of unused medications (Φc=0.134, p=0.014).

### Association between the storage and disposal practices of expired medications and the socio-demographic characteristics

As shown in Table 5, educational status was moderately associated with possession of expired medication at home (Φc=0.254, p<0.001), checking the expiration date before purchasing (Φc=0.297, p<0.001), and checking the expiration date before using medications (Φc=0.251, p=0.001), but was weakly associated with the proper disposal method of expired medications (Φc=0.152, p=0.003). Monthly income was moderately associated with possession of expired medication at home (Φc=0.279, p<0.001) but was weakly associated with the proper disposal method of expired medications (Φc=0.171, p<0.001).

## Discussion

The very high level of unused medication storage observed in this study (97%) suggests that keeping medicines at home is a common household practice in this setting. To our knowledge, this proportion is higher than that reported in previous studies. For instance, studies conducted in Ethiopia reported prevalence rates of 41.4% [24] and 29% [25], while studies in Pakistan (35.3%) [26] and Tanzania (70.1%) [27] reported lower rates. The closest estimate was reported in Afghanistan (95.3%) [6]. Variations in awareness about the risks of unused medication storage in homes and the availability of a system for returning unused medications could explain the difference. The practice of storing unused medications at home could facilitate unintentional poisoning, particularly among children, as well as self-medication and medication misuse.

The prominence of emergency use as the main reason for retaining unused medications may reflect uncertainty about future access to treatment and a tendency to keep medicines for potential later need. Previous studies reported that improvement in health conditions, adverse drug reactions, and future use as the common reasons for unused medication storage [24].

**Table 5. Association between the storage and disposal practices of expired medications and the socio-demographic characteristics of the participants.**

| Storage and disposal practices of expired medications | Age group | Marital status | Education | Occupation | Monthly Income |
|---|---|---|---|---|---|
| Do you currently have expired medication at home? | Φc = 0.140 p = 0.103 | Φc = 0.080 p = 0.740 | **Φc = 0.254 p < 0.001\*** | Φc = 0.110 p = 0.071 | **Φc = 0.279 p < 0.001\*** |
| Why do you keep expired medication? | Φc = 0.203 p = 0.120 | Φc = 0.178 p = 0.331 | Φc = 0.214 p = 0.406 | Φc = 0.314 p = 0.372 | Φc = 0.419 p = 0.213 |
| Do you check the expiration date before purchasing a medication? | Φc = 0.128 p = 0.072 | Φc = 0.068 p = 0.158 | **Φc = 0.297 p < 0.001\*** | Φc = 0.046 p = 0.353 | Φc = 0.390 p = 0.053 |
| Do you check the expiration date before using a medication? | Φc = 0.134 p = 0.052 | Φc = 0.073 p = 0.129 | **Φc = 0.251 p = 0.001\*** | Φc = 0.017 p = 0.731 | Φc = 0.379 p = 0.071 |
| Are you familiar with the proper method for disposing of a nearly expired medication? | Φc = 0.125 p = 0.084 | Φc = 0.060 p = 0.213 | Φc = 0.110 p = 0.272 | Φc = 0.037 p = 0.448 | Φc = 0.151 p = 0.046 |
| What method do you consider appropriate for disposing of a nearly expired medication? | Φc = 0.100 p = 0.615 | Φc = 0.053 p = 0.945 | Φc = 0.131 p = 0.078 | Φc = 0.145 p = 0.109 | Φc = 0.101 p = 0.615 |
| Are you familiar with the proper method for disposing of an expired medication? | Φc = 0.075 p = 0.489 | Φc = 0.026 p = 0.598 | Φc = 0.113 p = 0.242 | Φc = 0.027 p = 0.582 | Φc = 0.101 p = 0.358 |
| What is the proper method for disposing of an expired medication? | Φc = 0.081 p = 0.501 | Φc = 0.103 p = 0.209 | **Φc = 0.152 p = 0.003\*** | Φc = 0.055 p = 0.735 | **Φc = 0.171 p < 0.001\*** |

**Note:** Chi-square test was used to assess associations. Φc represents Cramer's V effect size. **\*p < 0.05** indicates statistical significance.

Medications were commonly kept in bedroom storage spaces, a practice that may increase accessibility within the home and raise safety concerns, particularly for children. This finding is in line with other studies carried out in Tanzania [27], Ethiopia [25], and Palestine [28]. Improper medication storage locations can lead to accidental poisoning, particularly among children [25].

The marked gap between self-reported familiarity with proper medication storage and the very limited use of medication label instructions as a source of guidance suggests that participants' knowledge may rely more on informal understanding than on written guidance. In the present study, 98% of respondents did not read medication storage instructions, which is substantially higher than the proportion reported in a study conducted in Pakistan, where only 20.9% of respondents did not read storage instructions [18]. This difference may be explained by variations in educational level and awareness.

The presence of expired medications in households (34%) indicates an important medicine safety concern and suggests gaps in routine medicine review and disposal practices. This finding is higher than the rates reported across studies conducted in European countries such as Serbia (10.3%) [29], Croatia (10.8%) [30], and Belgium (21%) [31], as well as studies conducted in Brazil (12.2%) [32] and the USA (23%) [33]. The higher percentage observed in this study may be attributed to differences in education, awareness, cultural practices, public health policies, and the availability of medication take-back systems.

The main reasons for keeping expired medications at home were forgetfulness and the intention to dispose of them later, suggesting that expired medications may remain in households passively rather than intentionally. Poor compliance, inappropriate prescribing, lack of regulation of prescription drug sales in community pharmacies, lack of awareness programs on proper medication storage and disposal, and the complete absence of a drug take-back system in the country could all be contributing factors to the possession of expired medications. Children are particularly vulnerable to unintentional ingestion of unused or expired medications kept at home [33].

Checking expiry dates before purchasing medications was uncommon (30%), indicating limited attention to basic medication safety practices. This finding is lower than that reported in studies conducted in Afghanistan (97%) [6] and Pakistan

(94.9%) [18], but higher than that reported in a study conducted in India (23.6%) [34]. Additionally, only 26% of participants checked the expiration date before using medications, which is lower than that reported in a study carried out in Pakistan (48.7%) [18].

Although most participants reported familiarity with proper disposal methods for unused, nearly expired, and expired medications, their actual practices suggested otherwise, because household trash was identified as the main disposal method. This inconsistency indicates that self-reported knowledge may reflect misconceptions rather than a correct understanding of recommended disposal practices, revealing a clear gap between perceived awareness and actual knowledge of proper medication disposal.

The predominance of household trash disposal is consistent with findings from other studies [6,35], while only a very small proportion of participants reported returning medications to health care facilities. The continued reliance on household disposal and the minimal use of facility-based return options may reflect the absence of structured take-back systems and inadequate public guidance on proper medication disposal. Similar patterns have been reported in studies from Afghanistan, Pakistan, South Africa [6,12,18], and Ethiopia [24]. In contrast, studies conducted in developed countries such as Portugal [36], New Zealand [19], and Sweden [37] have documented higher levels of returning unused medications to health care facilities. These differences may be partly explained by variations in public awareness, as poor disposal practices have been associated with limited knowledge of proper medication disposal [6,18,27,38].

Educational status was significantly associated with possession of unused and expired medications at home. Similar results were found in studies carried out in Jordan [39] and Palestine [28]. This may be attributed to the role of education in developing appropriate storage, use, and disposal of medications [40].

Income was another key socio-demographic variable associated with the possession of unused and expired medication at home. A study carried out in Qatar revealed that participants with higher incomes were less likely to keep medications at home [41]. Conversely, studies carried out in Uganda and Ethiopia revealed that participants with regular or higher incomes were more likely to store medications at home [42,43]. This could be attributed to the contextual differences in terms of access to medications, health insurance coverage, and cultural practices.

Although these findings provide useful insight into household medication storage and disposal practices, they should be interpreted with caution because the study used consecutive sampling and was conducted among women attending a single hospital outpatient pharmacy in Mogadishu, which may limit their applicability to other populations and settings in Somalia.

The findings of this study should be interpreted within the broader context of Somalia's health system and pharmaceutical governance environment. Recent reports indicate that the Somali health sector continues to face major structural challenges, including weak regulatory oversight, fragmented service delivery, and heavy reliance on private health providers and pharmacies for access to medicines. According to the Somali Ministry of Health, a substantial portion of the health system operates outside strong government regulation, while the private sector provides a large share of healthcare services in the country [44]. In addition, World Health Organization reports have emphasized the need to strengthen national pharmaceutical governance and improve access to essential medicines through regulatory reforms and capacity building within the medicines sector [45]. More recently, Somalia established the Interim National Medicines Regulatory Authority to enhance oversight of medical products and address longstanding challenges related to medicine quality and regulation [46]. Despite these ongoing reforms, empirical evidence on household-level practices related to medication storage and disposal in Somalia remains very limited.

## Limitations and strengths

This research represents the first study assessing medication storage and disposal practices in Somalia, thereby addressing an important knowledge gap in a fragile healthcare setting. The study highlights the lack of medication take-back systems and public awareness programs and provides practical recommendations to improve medication

storage and disposal practices. However, several limitations should be considered. The cross-sectional design does not allow causal inference. In addition, the study used consecutive sampling and was conducted at a single hospital outpatient pharmacy in Mogadishu, which may introduce selection bias and limit the generalizability of the findings to other populations, healthcare settings, and regions of Somalia. Because the study included only adult women attending this facility, the findings may not fully represent medication storage and disposal practices in the wider community.

## Conclusion

The study found a high prevalence of unused medications stored at home, primarily for emergencies, with the bedroom cabinet being the main storage location. Most participants did not read medication storage instructions and reported disposing of medications in household trash. More than one-third of participants also kept expired medications at home, mainly due to forgetfulness. In addition, education and income were significantly associated with several medication storage and disposal practices. The findings suggest the need for greater public awareness of proper medication storage and disposal practices. They also support consideration of strategies to encourage the return of unused and expired medications to pharmacies, as well as the development of a drug take-back system and national guidance on proper medication storage and disposal.

## Supporting information

**S1_File. Questionnaire.** This file contains all the questions used for this study.
(PDF)

**S2_File. Dataset.** This file contains the complete dataset necessary to replicate the statistical analyses and findings reported in this study.
(XLSX)

## Acknowledgments

We express our genuine gratitude to the participants for their valuable contribution to the study. We would like to express our gratitude to Banadir Hospital for granting us permission to collect data from participants. Last but not least, we would like to acknowledge the Research Ethics Committee (REC) of Al Hayat Medical University for providing ethical approval for this study.

## Author contributions

**Conceptualization:** Abdiweli Mohamed Abdi.

**Data curation:** Abdinor Hussein Ahmed, Mohamed Yusuf Isak.

**Formal analysis:** Abdiweli Mohamed Abdi.

**Methodology:** Osman Mohamed Mohamud, Nor Haji Osman.

**Project administration:** Ahmed Abdulkadir Kalif, Mahad Ali Mohamud.

**Resources:** Osman Mohamed Mohamud, Nor Haji Osman.

**Supervision:** Abdinor Hussein Ahmed, Mohamed Yusuf Isak.

**Visualization:** Ahmed Abdulkadir Kalif, Mahad Ali Mohamud.

**Writing – original draft:** Abdiweli Mohamed Abdi.

**Writing – review & editing:** Mohamed Ismail Iman.

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
