## [Decision Letter · Decision Letter 0]

10 Mar 2026

Dear Dr. Abdi,

Thank you for submitting your manuscript to PLOS ONE. After careful consideration, we feel that it has merit but does not fully meet PLOS ONE’s publication criteria as it currently stands. Therefore, we invite you to submit a revised version of the manuscript that addresses the points raised during the review process.

We look forward to receiving your revised manuscript.

Kind regards,

Jamil Afzal, Ph.D, Post Doc

Academic Editor

PLOS One

Journal Requirements:

2. In the ethics statement in the Methods, you have specified that verbal consent was obtained. Please provide additional details regarding how this consent was documented and witnessed, and state whether this was approved by the IRB

“None”

4. We note that your Data Availability Statement is currently as follows: “All relevant data are within the manuscript and its Supporting Information files.”

Additional Editor Comments:

• The manuscript should better justify why this sampling method was selected, discuss potential selection bias, and explain why it only included individuals visiting the hospital pharmacy.

• Revise the discussion part; it would benefit from a deeper exploration of contextual factors specific to Somalia.

• Provide additional information on questionnaire validation; the complete questionnaire should be included in an appendix to improve clarity and transparency

Reviewers' comments:

Reviewer's Responses to Questions

**Comments to the Author**

1. Is the manuscript technically sound, and do the data support the conclusions?

Reviewer #1: Partly

Reviewer #2: Yes

2. Has the statistical analysis been performed appropriately and rigorously?

Reviewer #1: Yes

Reviewer #2: I Don't Know

3. Have the authors made all data underlying the findings in their manuscript fully available?

Reviewer #1: Yes

Reviewer #2: Yes

4. Is the manuscript presented in an intelligible fashion and written in standard English?

Reviewer #1: Yes

Reviewer #2: Yes

Reviewer #1: This manuscript presents a cross-sectional study on the storage and disposal practices of unused and expired medications among adult women attending Banadir Hospital in Mogadishu, Somalia. The topic is relevant from a public health perspective, particularly in low-resource and fragile health system settings, and the study addresses a clear gap in the literature, as data from Somalia are currently very limited. Overall, the manuscript is well organized, and the objectives, methods, and main findings are clearly presented. But many points that requires clarification .

1-The abstract contains a large amount of numerical detail, which makes it dense. Streamlining the abstract to highlight the most important findings and implications would improve clarity.

2- The study uses consecutive sampling at a single hospital outpatient pharmacy. While this is reasonable in the given context, it limits the generalizability of the findings. This should be more clearly acknowledged in the Discussion and Limitations sections, and conclusions should be phrased cautiously.

3-The manuscript frequently refers to “proper” storage and disposal practices; however, Somalia currently lacks national guidelines for medication disposal. The authors should clearly define what they consider “proper” practices and indicate whether these are based on WHO or other international recommendations.

4-A large proportion of participants reported being familiar with proper disposal methods, yet most identified household trash as the correct method. This apparent inconsistency suggests misconceptions rather than true knowledge and should be explicitly discussed.

5-More information is needed on how key terms such as “unused medication” and “expired medication” were explained to participants during the interviews, to ensure consistent understanding.

6-The use of chi-square tests and Cramer’s V is appropriate. However, the manuscript describes some associations as “strong” or “moderate” without specifying the thresholds used. The authors should define these categories or use more neutral wording.

7-The Discussion section is well referenced but occasionally repeats results already presented earlier. Reducing repetition and focusing more on interpretation and implications would strengthen this section.

8-The manuscript would benefit from careful language and formatting revision to address minor grammatical errors, spacing issues, and inconsistencies throughout the text.

Reviewer #2: Overall, the manuscript is well written and clearly structured. The study appears to be methodologically sound, and the authors demonstrate careful attention to detail, particularly in the data collection process, which supports the reliability of the findings.

Furthermore, the study addresses an important issue within the local context. The findings may provide useful insights for the development of targeted programs or awareness initiatives for the relevant community groups.

.

Reviewer #1: No

Reviewer #2: No

---

## [Author Response · Author response to Decision Letter 1]

18 Mar 2026

Dear Editor,

Thank you for the opportunity to revise our manuscript entitled “Storage and disposal practices of unused and expired medications among adult women attending Banadir Hospital, Mogadishu, Somalia: A cross-sectional study.”

We have carefully revised the manuscript in response to the academic editor’s and reviewers’ comments. A detailed point-by-point response to all comments has been uploaded as a separate file labeled Response to Reviewers. We have also uploaded a marked-up version of the manuscript as Revised Manuscript with Track Changes and a clean revised version as Manuscript.

In summary, we revised the manuscript to align with PLOS ONE formatting requirements, clarified the ethics statement and verbal consent procedure, updated the Data Availability Statement, expanded the justification for the sampling approach and its limitations, strengthened the discussion of Somalia-specific contextual factors, provided additional information on questionnaire validation, included the complete questionnaire as Supporting Information, clarified key definitions used during interviews, defined the thresholds used for interpreting Cramér’s V, reduced repetition in the Discussion, and carefully revised the language and formatting throughout the manuscript.

We also confirm that the correct Funding Statement is: “The authors received no specific funding for this work.”

We hope that the revised manuscript is now suitable for publication in PLOS ONE.

Sincerely,

Abdiweli Mohamed Abdi

Submitting Author

---

## [Decision Letter · Decision Letter 1]

16 Apr 2026

Storage and disposal practices of unused and expired medications among adult women attending Banadir Hospital, Mogadishu, Somalia: A cross-sectional study

PONE-D-26-01866R1

Dear Dr. Abdi,

We’re pleased to inform you that your manuscript has been judged scientifically suitable for publication and will be formally accepted for publication once it meets all outstanding technical requirements.

Kind regards,

Jamil Afzal, Ph.D, Post Doc

Academic Editor

PLOS One

Additional Editor Comments (optional):

Reviewers' comments:

Reviewer's Responses to Questions

**Comments to the Author**

Reviewer #1: All comments have been addressed

Reviewer #2: All comments have been addressed

2. Is the manuscript technically sound, and do the data support the conclusions?

Reviewer #1: Yes

Reviewer #2: Yes

3. Has the statistical analysis been performed appropriately and rigorously?

Reviewer #1: Yes

Reviewer #2: I Don't Know

4. Have the authors made all data underlying the findings in their manuscript fully available?

Reviewer #1: Yes

Reviewer #2: Yes

5. Is the manuscript presented in an intelligible fashion and written in standard English?

Reviewer #1: Yes

Reviewer #2: Yes

Reviewer #1: The authors have adequately addressed all comments from the previous review. The study is well-designed and the methodology is appropriate for the research question. Data are clearly presented and support the conclusions. Statistical analyses are rigorous, and the manuscript is written clearly in standard English. Overall, the manuscript is scientifically sound and can be accepted for publication

Reviewer #2: (No Response)

.

Reviewer #1: **Yes:**Yes, I am happy for my identity to be disclosed for this peer reviewYes, I am happy for my identity to be disclosed for this peer reviewYes, I am happy for my identity to be disclosed for this peer reviewYes, I am happy for my identity to be disclosed for this peer review

Reviewer #2: No

---

## [Editor Report · Acceptance letter]

PONE-D-26-01866R1

PLOS One

Dear Dr. Abdi,

I'm pleased to inform you that your manuscript has been deemed suitable for publication in PLOS One. Congratulations! Your manuscript is now being handed over to our production team.

Kind regards,

on behalf of

Dr. Jamil Afzal

Academic Editor

PLOS One